# Engineered rHDL Nanoparticles as a Suitable Platform for Theranostic Applications

**DOI:** 10.3390/molecules27207046

**Published:** 2022-10-19

**Authors:** Liliana Aranda-Lara, Keila Isaac-Olivé, Blanca Ocampo-García, Guillermina Ferro-Flores, Carlos González-Romero, Alfredo Mercado-López, Rodrigo García-Marín, Clara Santos-Cuevas, José A. Estrada, Enrique Morales-Avila

**Affiliations:** 1Faculty of Medicine, Universidad Autónoma del Estado de México, Toluca 50180, Estado de México, Mexico; 2Department of Radioactive Materials, Instituto Nacional de Investigaciones Nucleares, Ocoyoacac 52750, Estado de México, Mexico; 3Faculty of Chemistry, Universidad Autónoma del Estado de México, Toluca 50120, Estado de México, Mexico

**Keywords:** radiolabeled nanoparticles, lutetium-nanoparticles, technetium-nanoparticles, high-density lipoproteins, theranostics, drug delivery systems, apolipoprotein A1

## Abstract

Reconstituted high-density lipoproteins (rHDLs) can transport and specifically release drugs and imaging agents, mediated by the Scavenger Receptor Type B1 (SR-B1) present in a wide variety of tumor cells, providing convenient platforms for developing theranostic systems. Usually, phospholipids or Apo-A1 lipoproteins on the particle surfaces are the motifs used to conjugate molecules for the multifunctional purposes of the rHDL nanoparticles. Cholesterol has been less addressed as a region to bind molecules or functional groups to the rHDL surface. To maximize the efficacy and improve the radiolabeling of rHDL theranostic systems, we synthesized compounds with bifunctional agents covalently linked to cholesterol. This strategy means that the radionuclide was bound to the surface, while the therapeutic agent was encapsulated in the lipophilic core. In this research, HYNIC-S-(CH_2_)_3_-S-Cholesterol and DOTA-benzene-*p*-SC-NH-(CH_2_)_2_-NH-Cholesterol derivatives were synthesized to prepare nanoparticles (NPs) of HYNIC-rHDL and DOTA-rHDL, which can subsequently be linked to radionuclides for SPECT/PET imaging or targeted radiotherapy. HYNIC is used to complexing ^99m^Tc and DOTA for labeling molecules with ^111, 113m^In, ^67, 68^Ga, ^177^Lu, ^161^Tb, ^225^Ac, and ^64^Cu, among others. In vitro studies showed that the NPs of HYNIC-rHDL and DOTA-rHDL maintain specific recognition by SR-B1 and the ability to internalize and release, in the cytosol of cancer cells, the molecules carried in their core. The biodistribution in mice showed a similar behavior between rHDL (without surface modification) and HYNIC-rHDL, while DOTA-rHDL exhibited a different biodistribution pattern due to the significant reduction in the lipophilicity of the modified cholesterol molecule. Both systems demonstrated characteristics for the development of suitable theranostic platforms for personalized cancer treatment.

## 1. Introduction

The possibility of identifying and recording images of molecular events in tissues or cells depends on the specific recognition and accumulation of the probe. Since the therapeutic response can be molecularly monitored in real-time, the co-administration of diagnostic probes and therapeutic agents on a single platform offers advantages in the research of new drugs for clinical applications. Theranostic systems (combined therapy–imaging systems) based on carrier nanoparticles (NPs) are used in several diseases, although they are especially valuable in personalized cancer therapy. Nanomedicine’s growth made possible the development of theranostic systems mainly based on inorganic NPs (Fe_x_O_y_, Au, SiO_2_, etc.) and organic materials (polymers, lipids, surfactants, etc.), which transport and release therapeutic and/or imaging agents, including radionuclides, at specific molecular sites. Reconstituted high-density lipoproteins (rHDLs) have physicochemical and biological properties like those of endogenous HDLs. rHDLs have a multimodal carrying capacity since they can transport imaging agents and specifically release drugs in cancer cells concomitantly [1,2,3,4,5,6,7,8,9,10]. The endogenous HDL uptake is driven by cell receptors, specifically by the SR-B1 protein expressed in the liver, adipose tissue, and steroidogenic organs. Besides, SR-B1 has been widely identified in a great variety of tumor cells, associated with an increase in cholesterol metabolism [5,9,10,11,12,13].

The idea for the use of rHDL as a drug delivery system is widely explored. It is possible to transport ions and molecules into rHDL both in the inner core and/or on the particle surface [2,5,7,8,9,14]. Hydrophobic molecules can be inserted into the inner core, which is mainly composed of cholesterol, cholesterol esters, and triglycerides, while hydrophilic molecules can be absorbed or conjugated to surface molecules to allow the hydrophilic interaction with the aqueous medium [2,4,6,15].

Several strategies for nanoparticle surface modification have been reported, such as (a) the covalent bonding to phospholipids (PLs) or Apo A1 on the surface of a bifunctional agent (BFA), which subsequently couples to the ion or molecule to be inserted, and (b) the non-covalent intercalation within surface phospholipids of molecules interacting through ionic forces, electrostatic interactions, or hydrogen bonds with the molecules to be linked [2,4,8,15,16,17]. It is important to note that the loading on the system and the surface-performed modifications can compromise the in vivo behavior of the NPs, in particular the specific recognition by the SR-B1.

The preparation of the nano-theranostic rHDL platform includes loading with therapeutic agents in the nanoparticle core. This idea is based on two important premises: (a) most anticancer drugs are poorly soluble in water, so they can be easily incorporated into the rHDL core [3,6,10,18], and (b) once rHDL binds to the SR-B1 present on the surface of the tumor cell, it contributes to the release of hydrophobic substances directly into the cytosol, thus increasing significantly the therapeutic efficacy [9,10,19].

Since the function of gamma-emitting radionuclides is to provide the rHDL imaging at the site of cargo delivery, they can be attached on the nanoparticle surface. It has been reported that the cell internalization of drugs released from the rHDL are not efficient when the cargo is on the nanoparticle surface; therefore, drugs have to be loaded in the rHDL core [20,21]. Therefore, the complexing of radionuclides to BFA attached to rHDL through cholesterol structures would enable the theranostic capabilities of rHDL nanoparticles. 6-hidrazinylnicotinic acid (HYNIC) is a BFA widely used in the preparation of ^99m^Tc-radiopharmaceuticals. DOTA (2,2′,2″,2‴-(1,4,7,10-tetraazacyclododecane-1,4,7,10-tetrayl)tetraacetic acid) is a BFA very efficient as a chelator of diagnostic and therapeutic radionuclides, such as ^111^In, ^64^,^67^Cu, ^67^,^68^Ga, ^161^Tb, ^225^Ac, and ^177^Lu [22,23].

The objective of this research was to prepare surface-modified rHDL to anchor radionuclides useful for molecular imaging. Therefore, this nanoplatform will have theranostic properties since the rHDL core will be available for the transport of therapeutic agents. Two types of modified rHDL were prepared, one with HYNIC and the other with DOTA as BFAs. For this purpose, BFAs were conjugated to cholesterol molecules, and with these cholesterol derivatives, the rHDL NPs were synthesized.

## 2. Results

The preparation was divided into three stages. In the first one, the derivatives HYNIC-S-(CH_2_)_3_-S-Cholesterol (HYNIC-Cholesterol) and DOTA-benzene-p-SC-NH-(CH_2_)_2_-NH-Cholesterol (HYNIC-Cholesterol) were synthesized (see Figure 1a). In the second one, rHDL was prepared, using the HYNIC-Cholesterol and DOTA-Cholesterol as components in the formulation (Figure 1b). In the third stage, the NPs of HYNIC-rHDL were radiolabeled with ^99m^Tc and that of DOTA-rHDL with ^177^Lu (Figure 1b).

### 2.1. Preparation and Characterization of HYNIC-Cholesterol and DOTA-Cholesterol Derivatives

The preparation of HYNIC-Cholesterol and DOTA-Cholesterol was carried out in three steps. In the first, a halogenated cholesteryl derivative was prepared; posteriorly, Cholesterol was modified with a crosslinking reagent to obtain the -SH or -NH_2_ ending, and finally, the chelating agent HYNIC or DOTA was conjugated. Table 1 shows the results of the reaction yield, physical appearance, melting point, and Rf by the TLC-SG of HYNIC-Cholesterol, DOTA-Cholesterol, and precursor species involved in their preparation.

The Rf values and UV-Vis spectra of HYNIC-Cholesterol, DOTA-Cholesterol, and precursor species showed significant changes during the consecutive reactions, indicating the formation of new compounds (Figure 2). The cholesterol spectrum (Rf = 0.60) is characterized by a wide absorption band between 225 and 260 nm [24]. The cholesterol-derivatives (Br-/Cl-) (Rf = 0.57/0.45) showed the loss of the absorption band. When the thiolate-Cholesterol was formed (Rf = 0.74), a slight absorption band (shoulder) near 280 nm suggested the presence of -SH groups. The spectrum of the NH_2_-Cholesterol derivative (Rf = 0.83) showed, unlike its precursor chlorinated cholesterol (Rf = 0.45), a band at 229 nm, which was also different from the 233 nm band of the original cholesterol. This band (229 nm) indicates the presence of the -NH_2_ group due to n→ σ* transitions, which acts as an auxochrome with absorbance bands close to 220 nm. Finally, the HYNIC-Cholesterol compound (Rf = 0.67) showed a characteristic absorption band centered at 277 nm. The DOTA-Cholesterol compound (Rf = 0.83) shows characteristic bands at 265 nm and 279 nm, corresponding to the π → π* transitions of the partial resonant character of the carbonyl groups from the DOTA molecule.

The infrared (IR) spectra are shown in Figure 3. The characteristic bands of cholesterol appeared in the spectra of all derived compounds. The spectrum of Br-Cholesterol showed a distinctive band at 598 cm^−1^, which corresponds to (C-Br)_υ_. The Cl-Cholesterol spectrum obtained coincided with that reported by NIST. This spectrum showed a weak band at 440 cm^−1^, which corresponds to the 280–500 cm^−1^ region characteristic of chloroalkanes (CCl)_δ_. In addition, the 3423 cm^−1^ band characteristic of the -OH group in the cholesterol spectrum was absent, which suggests that chlorine replaced the -OH in the cholesterol molecule.

In the IR spectrum of SH-(CH_2_)_3_-SH-Cholesterol (SH-Cholesterol) (Figure 3), the band at 835 cm^−1^ corresponds to (S-H)_δ_ and the band at 630 cm^−1^ corresponds to stretching (C-S)_υ_, while the (C-Br)_υ_ band disappeared. The spectrum of NH_2_-(CH_2_)_2_-NH-Cholesterol (NH_2_-Cholesterol) (Figure 3) showed a broad band at 3400–3500 cm^−1^, which characterizes the presence of a primary amine (NH_2_)_st_; the disappearance of the vibration associated with the chloroalkane (CCl)_δ_ was also observed, from which it was inferred that the chlorine was substituted. The spectrum of DOTA-Cholesterol showed common bands (although shifted) with regard to the DOTA-benzene*-p-*SCN spectrum.

In the HYNIC-Cholesterol spectrum (Figure 3), a characteristic weak band at 3386 cm^−1^ was observed, corresponding to the (N-H)_υ_ stretch of HYNIC. A significant modification was identified from vibration corresponding to the (C=O)_υ_ of the carbonyl group at 1785 and 1729 cm^−1^, which was displaced due to the high energy attributed to the pyridine ring and the sulfur group. The bands found at 1458 and 1378 cm^−1^ correspond to the pyridine skeleton.

The DOTA-Cholesterol spectrum (Figure 3) showed common bands (although shifted) compared with those of the raw material (cholesterol) spectrum. The band of middle intensity at 1219 cm^−1^ (in DOTA-benzene*-p-*SCN at 1193 cm^−1^) was attributed to the presence of the (C=S)_st_ bond, which is part of the thiocarbonyl group, which binds both molecules. In addition, the band at 2049 cm^−1^ in the DOTA-Cholesterol spectrum indicated that the aromatic group was still present in the molecule.

### 2.2. Preparation of HYNIC-rHDL and DOTA-rHDL Nanoparticles

HYNIC-rHDL and DOTA-rHDL NPs were prepared by single emulsion, resulting in particle sizes of 60 ± 13 nm (PDI: 0.85) and 75.12 ± 12 nm (PDI: 0.97) respectively; the Zeta potential of the rHDL systems was superior to +18 mV. These particle sizes had a variation of less than 5% during the following 48 h. For the in vitro evaluation, rHDL NPs without surface modification (76.60 ± 10 nm) were used as a control. The size of all samples was adequate to avoid phagocytosis by the RES and enter the tumor microenvironment through the fenestrated microvasculature [25].

### 2.3. Radiolabeling of HYNIC-rHDL and DOTA-rHDL Nanoparticles

The radiochemical yield was 55% and 85% for ^99m^Tc-HYNIC-rHDL NPs and ^177^Lu-DOTA-rHDL NPs, respectively, where the main impurities were ^99m^TcO_4_Na and ^177^LuCl_3_. Radiolabeled nanoparticles were purified by centrifugation using a 100,000 MWCO membrane tube. The radiochemical purity of the final product in both preparations was greater than 99%, as determined by size-exclusion HPLC chromatography (Figure 4). Furthermore, the ^177^Lu- and ^99m^Tc-labeled rHDL nanosystems showed high stability in saline and human serum (radiochemical purity > 95% assessed after 24 h of incubation at 37 °C by size-exclusion HPLC). For comparative purposes, ^99m^Tc-HYNIC-Cholesterol and ^177^Lu-DOTA-Cholesterol were prepared with radiochemical yields and purities higher than 99% (Figure 4). A pronounced shift of the rHDL nanoparticles (both radiolabeled and unlabeled) to a higher molecular weight (T_R_ = 4.7–5.5 min) with regard to their cholesterol precursors (T_R_ = 8.8–10.1 min) was clearly observed (Figure 4). These results proved that the cholesterol derivatives conjugated to the different BFAs were effectively incorporated during the formation of the rHDL nanoparticles (protein nature due to the Apo-A1 incorporation). It is important to point out that the relatively low radiolabeling yield of the rHDL nanoparticles with regard to their cholesterol derivatives was due to the fact that rHDL cannot be exposed to temperatures above 40 °C during the labeling procedure to avoid Apo-A1 denaturation with the consequent loss of the nanosystem recognition by SR-B1. Nevertheless, the final radiochemical purity of the radiolabeled rHDL nanoparticles is suitable for application in humans.

Figure 5 shows the in vitro results of HYNIC-rHDL(R6G) and DOTA-rHDL(R6G) interaction with T47D cells (hormone-dependent luminal subtype A human breast cancer with high expression of SR-B1) obtained by fluorescence microscopy. Rhodamine 6G (R6G) was used as a rHDL cargo to demonstrate the R6G internalization and release into the cytosol [26]. The expression of SR-B1 in T47D cells by the Western blot assay was previously determined [27]. The figure also shows that R6G internalization decreased after pretreatment with Apo-A1 in the form of empty rHDL (SR-B1 receptor blocking effect). The behavior of HYNIC-rHDL(R6G) and DOTA-rHDL(R6G) in cells indicated that the functionalization of the HYNIC and DOTA groups on the surface of rHDL did not affect its specific recognition or the ability of rHDL NPs to release and internalize their content (cargo).

#### 2.3.1. In Vitro Uptake of ^99m^Tc-HYNIC-rHDL and ^177^Lu-DOTA-rHDL

As expected, the data showed a low uptake and internalization (cytosol accumulation) for both rHDL radio-nanosystems in PC3 cells (SR-B1+) (Figure 6). The rHDL cargo release mechanism, previously reported, indicated that once the nanoparticles were recognized by the SR-B1 receptor, they released their lipid core load, followed by uncoupling from the receptor and returning to the bloodstream [3]. Therefore, rHDL NPs did not accumulate in the membrane. The results showed the cellular uptake in the membrane in a moment; therefore, its value was low (Figure 6). Nevertheless, this uptake was specific as it was lower in the presence of a blocking effect.

The rHDL NPs evaluated in this work were modified on their surface, but they were not loaded in the lipid core with any radioactive compound, so they had no radioactive cargo to release or accumulate in the cytosol. Hence, their internalization was very low and occurred by passive and non-specific diffusion, which was corroborated by the fact that there was no internalization variation when the cells were partially blocked. In the work of Isaac-Olive et. al. [7], rHDL showed high internalization of ^99m^Tc-HYNIC-DA, but it was loaded in the lipid core of the nanoparticle. Therefore, the results obtained showed that the recognition of rHDL NPs modified on their surface with HYNIC and DOTA remained specific.

#### 2.3.2. In Vivo Evaluation of the HYNIC-rHDL and DOTA-rHDL Systems

Figure 7 shows the in vivo behavior of the ^99m^Tc-HYNIC-rHDL and ^177^Lu-DOTA-rHDL nanoparticles at 2 and 24 h after of administration. Consistent with the basis of radiotracers, the mass amount of ^99m^Tc and ^177^Lu is a few atoms, so, after decay corrections (Figure 7), the biodistribution represents the in vivo biological pattern of HYNIC-rHDL and DOTA-rHDL. It was observed that the distribution in mice showed a similar behavior between the rHDL (without surface modification) [7] and HYNIC-rHDL, while the DOTA conjugation induced greater behavioral changes in the biodistribution profile with regard to rHDL modified with HYNIC. At 2 h post-injection, the DOTA-rHDL accumulation in kidneys was significantly greater (eight-times) than that of HYNIC-rHDL. HYNIC-rHDL also showed a greater liver uptake with respect to rHDL [7]. At the same distribution times, the accumulation of DOTA-rHDL in liver and spleen was lower than those of rHDL and HYNIC-rHDL (Figure 7).

The biodistribution pattern differences between HYNIC-rHDL and DOTA-rHDL were corroborated with the molecular imaging obtained at 2 h after administration of each nanosystem, radioisotopic image for ^99m^Tc-HYNIC-rHDL and optical image (Cerenkov image) for ^177^Lu-DOTA-rHDL (Figure 8).

## 3. Discussion

To bind molecules covalently to the surface of rHDL, the most common pathways are (a) binding of Apo-A1 to lysine, arginine, tyrosine, or cysteine residues [2,9,14,15,16,17] and (b) the binding to the head groups of the phospholipids [2,3,8,13,14,16,17,28]. The binding of molecules to cholesterol has been less exploited, although it has been used to carry siRNA [4,29], Gd chelates, and fluorescent molecules [14] to the surface of rHDL.

Cholesterol can be introduced in high proportions into the surface monolayer of rHDL during its preparation [11]. In this research, we demonstrated that if rHDL is prepared with an amphiphilic derivative of cholesterol synthesized with a bifunctional agent, where most of the hydrophobic domine of the molecule was “embedded” inside the lipidic nucleus, the hydrophilic part (which includes the bifunctional agent) is exposed on the outside, which makes it possible to successfully radiolabel the nanosystem with ^99m^Tc or ^177^Lu through the BFA exposed on the nanoparticle surface (Figure 1). Furthermore, this methodology allowed maintaining the Apo-A1 affinity, a critical point for the stability and rHDL recognition by cell receptors (SR-B1) [4,9,10,12].

To bind HYNIC and DOTA to cholesterol, the derivatives HYNIC-S-(CH_2_)_3_-S-Cholesterol and DOTA-benzene-p-SC-NH-(CH_2_)_2_-NH-Cholesterol were synthesized. Both syntheses required the modification of Cholesterol. Due to its central sterol nucleus, a suitable alternative to the modification is the substitution of the hydroxyl side group; the cholesterol molecules were halogenated (Br- or Cl-) and posteriorly modified using a cross-linker, which performs a spacing function and provides a functional group that facilitates later conjugation with HYNIC (thioester bond) or DOTA (thiourea). The synthesis of both derivatives was based on relatively easy methods, since this was based on simple organic reactions, which makes it very reproducible and accessible to many laboratories.

The integration of cholesterol derivatives to the rHDL preparation was also easy, and it was demonstrated that the recognition of SR-B1 on T47D cells’ surface was not affected. In addition, the use of Rhodamine 6G as a model of therapeutic drug confined in the lipidic core confirmed the ability of HYNIC-rHDL and DOTA-rHDL to load, transport, and release molecules to the cytosol of cancer cells. When the SR-B1 was blocked, the internalization of rHDL(R6G) decreased, due to a competitive effect on receptors, proving that the cell recognition and internalization were mediated by a ligand–receptor mechanism, thus demonstrating thus rHDL nanoparticles can be used as vectorized nanoplatforms for the design of smart drug delivery systems.

Once having verified that the bifunctional agent does not significantly interfere with the recognition and internalization mechanisms, HYNIC-rHDL and DOTA-rHDL were radiolabeled with ^99m^Tc and ^177^Lu, respectively. HYNIC is unable to complete the ^99m^Tc coordination sphere, forcing the use of additional coligands, ethylenediamine-*N*,*N*’-diacetic acid (EDDA) and N-[2-hydroxy-1,1-bis(hydroxymethyl)ethyl]glycine (tricine). The EDDA-tricine mixture is very useful for ^99m^Tc labeling using HYNIC due to the high stability of the ^99m^Tc-HYNIC bond. The final ^99m^Tc-HYNIC/EDDA complex showed high stability in vivo [7,30,31].

The formation of the ^99m^Tc-EDDA/HYNIC and ^177^Lu-DOTA radiocomplexes generally takes place at temperatures above 90 °C for 15–30 min. Therefore, the relatively low radiolabeling yield of the rHDL nanoparticles was due to the fact that rHDL cannot be exposed to temperatures above 40 °C during the labeling procedure to avoid Apo-A1 denaturation, which consequently would produce a loss of the nanosystem recognition by SR-B1. Nevertheless, the final radiochemical purity of the radiolabeled rHDL nanoparticles after purification by ultracentrifugation was as high as 99%.

The lower radiolabeling yield of HYNIC-rHDL nanoparticles (55%) compared to DOTA-rHDL nanoparticles (85%) was attributed to differences in the chemical structure of both cholesterol derivatives, which may be influenced as follows: (a) HYNIC can only coordinate through the pyridine ring nitrogen and the -NH_2_ group from hydrazine. The phosphate groups of the phosphatidylcholine in rHDL strongly interact with amino groups, both by electrostatic interactions and the formation of hydrogen bond bridges [32,33]. It is highly probable that part of the -NH_2_ groups of the HYNIC-S-(CH_2_)_3_-S-Cholesterol react or interact with the phosphates from phosphatidylcholine, leaving a smaller fraction of them available for coordination with ^99m^Tc. The -NH groups of DOTA-benzene-p-SC-NH-(CH_2_)_2_-NH-Cholesterol can also react with phosphatidylcholine phosphates, but they are not involved in the subsequent coordination of ^177^Lu, so their interaction with phospholipids influenced to a lesser degree the radiolabeling reaction. In addition, the probability of forming hydrogen bond bridges and the influence on the coordination complex formation is higher in the -NH_2_ groups of HYNIC-S-(CH_2_)_3_-S-Cholesterol than in the -NH groups of DOTA-benzene-p-SC-NH-(CH_2_)_2_-NH-Cholesterol.

The N-terminal and C-terminal groups and polar side chains of Apo-A1 can also interact with the functional groups of HYNIC-S-(CH_2_)_3_-S-Cholesterol and DOTA-benzene-p-SC-NH-(CH_2_)_2_-NH-Cholesterol, which contributes to decreasing the total surface area available for coordination with radionuclides. In the case of HYNIC, as already described, the influence is greater than in DOTA. On the other hand, the lipids on the rHDL surface can diffuse towards the inner core as a function of several physicochemical parameters and plays an important role in the nanoparticle configuration (for example, the size, geometry, hydrophobicity, long-chain, saturation, etc., of the lipidic molecules) [34]. The estimation of the CLogP values using fragment-based prediction calculations (molinspiration and ChemDraw software) showed that HYNIC-Cholesterol is more hydrophobic (CLogP = 12.59) than DOTA-Cholesterol (CLogP = 3.83). This enables HYNIC-Cholesterol to better diffuse through the rHDL lipid layer and to be less exposed to the aqueous medium surrounding the nanoparticle, compared to DOTA-Cholesterol. As the hydrophilic part of the molecule is the one that binds to the radionuclide, present in the aqueous medium, the more “buried” the BFA is in the lipid layer, the lower the possibility of coordination with the radionuclide is.

When ^99m^Tc-HYNIC-rHDL and ^177^Lu-DOTA-rHDL systems were exposed to PC3 cells with partially blocked SR-B1, membrane uptake decreased due to the competitive mechanism. Cellular internalization was not affected with respect to the previously reported case of rHDL with ^99m^Tc transported in its core (^99m^Tc-HYNIC-DA) [7], because the radiolabeled modified nanosystems have no cargo to deliver. The results confirmed that HYNIC-rHDL and DOTA-rHDL NPs maintain the specific recognition by SR-B1 and that this receptor contributes to releasing the rHDL cargo directly into the cytosol.

Biodistribution studies of ^99m^Tc-HYNIC-rHDL and ^177^Lu-DOTA-rHDL with regard to rHDL without cholesterol-BFA [7] reflect the changes in biological behavior induced by the additional chemical groups in cholesterol. Depending on the size, shape, composition, and administration route, rHDL accumulates in the liver, kidneys, spleen, heart, aorta, lungs, bone marrow, and steroidogenic organs, mediated mainly by the expression of SR-B1 and its relative concentrations in these organs [4,7,8,10,12,16,17].

When the rHDL surface is modified or additional molecules are incorporated, changes in the in vivo behavior occur because physicochemical properties such as size, geometry, zeta potential, and surface charge, among others, alter biological interactions such as protein corona, plasma stability, and affinity to receptor binding [4,8,9,11,34,35,36,37,38,39,40]. Apo-A1 is responsible for the structural integrity and functionality of HDL.

Even though the composition of the lipids in rHDL does not disturb the global structure of the particle, the conformations that Apo-A1 adopts are influenced by different hydrophobic interactions, salt bridges, and van der Waals interactions, and consequently, its folding and local interactions vary and modify its biological behavior. Additionally, the lipid composition of HDL (including core lipids) has been shown to affect the Apo-A1 lipid–protein and protein–protein interactions in which the -N and -C terminal residues are involved, as well as the central region to that of the propeller; these changes in lipid–protein and protein–protein interactions affect the binding capacity of rHDL to cellular receptors and other ligands [11,14,34,35,37,38,39,40,41,42,43,44].

The biodistribution pattern of HYNIC-rHDL nanoparticles was like that of rHDL without surface modification. The main influence of HYNIC-Cholesterol was manifested in a higher uptake in the liver. The distribution pattern of the DOTA-rHDL nanoparticles indicated significant renal uptake and urine clearance, which did not occur with rHDL [7].

Other authors have also reported a significant increase in renal uptake of rHDL by modifying the nanoparticle surface [16,17]. Usually, the liver, skeletal muscle, adipose tissue, and certain cells such as macrophages are considered the primary sites of lipoprotein metabolism [45,46,47]. Although the liver is the main site of HDL uptake and degradation, the kidneys play an important role in its catabolism [48,49,50]. It is important to note that, in rodents, renal catabolism can be even more active than hepatic catabolism [51].

The metabolism of HDL appears more complex than other lipoproteins [52] because it involves the metabolism of several components, including Apo-A1 [45,46,47,53]. The rapid renal DOTA-rHDL clearance of nanoparticles suggests a greater interaction with plasma proteins, which may result in smaller particles [54], capable of being metabolized more efficiently in the kidney than HYNIC-rHDL particles. In addition, the individual components of rHDL (particularly Apo-A1) can be filtered, degraded, reabsorbed, and eliminated in the kidney independently [16,47,48,52], through different transporters and receptors, among which megalin, cubilin, and SR-B1 stand out [55].

The cells of the renal proximal tubes are the main site of Apo-A1/rHDL uptake in the kidney [17,56]. Megalin and cubilin receptors are presumed to have expressed themselves on these cells, forming a multi-receptor complex (megalin/cubilin) with a high affinity for Apo-A1/rHDL, contributing to its renal catabolism. Other mechanisms of HDL degradation have also been independent of the one indicated, but they are more linked to endogenous HDL, much more complex than rHDL [10,14,40,47,49,52].

In the renal metabolism of rHDL, the configuration and charge of the particle have a notable influence [56]. The different tertiary structures that Apo-A1 can adopt depending on the lipid composition [4,16,35,48,52].

It was shown that both HYNIC-rHDL and DOTA-rHDL are suitable for use as theranostic agents since they can carry molecules in their inner core and radionuclides on their surface without losing the specific recognition by the SR-B1 present in tumor cells.

## 4. Materials and Methods

### 4.1. Preparation of Derivatives HYNIC-S-(CH_2_)_3_-S-Cholesterol and DOTA-Benzene-p-SC-NH-(CH_2_)_2_-NH-Cholesterol

#### 4.1.1. Preparation of Halogen–Cholesterol Derivative

The preparation of the halogen–Cholesterol (X-Cholesterol) derivative can follow two alternatives, bromination or chlorination using phosphorus tribromide (PBr_3_) or thionyl chloride (SOCl_2_), respectively. Briefly, in a closed round flask with a condenser and anhydrous CaCl_2_ trap, a certain amount of cholesterol was dissolved in 5 mL of methylene chloride (DCM) by mixing. PBr_3_ or SOCl_2_ was then added in a 3:1 molar ratio to the cholesterol. The reaction mixture was kept under stirring for 3 h. After the reaction time, the mixture was transferred to a separatory funnel and then washed twice with distilled water. The final organic phase was separated and evaporated on a rotary evaporator and stored for posterior steps and characterization.

#### 4.1.2. Preparation of the -SH, -NH_2_ Derivatives: SH-(CH_2_)_3_-S-Cholesterol and NH_2_-(CH_2_)_2_-NH-Cholesterol

To obtain suitable reactive groups for the binding of chelating agents, two strategies were followed: the SH-(CH_2_)_3_-S-Cholesterol and NH_2_-(CH_2_)_2_-NH-Cholesterol were prepared using 1,3-dimercaptopropane (C_3_H_8_S_2_) and 1,2-diaminoethane (C_2_H_8_N_2_), respectively.

In a round flask, a certain amount of X-Cholesterol was dissolved in dimethylformamide (DMF) or acetonitrile (CH_3_CN), under constant stirring, in an ice bath. The 1,3-dimercaptopropane or 1,2-diaminoethane was added in a 1:1 molar ratio to X-Chol, and then, 2 equivalent moles of NaH were added. The reaction was under an inert atmosphere (N_2_), for approximately 3 h. After the reaction time, 3 mL of ethanol was added, followed by distilled water. The mixture was transferred to a separatory funnel, and posteriorly, a mixture of DCM: H_2_O (2:1) was added. The organic phase was separated and evaporated to dryness on a rotary evaporator. The product obtained was washed twice more with DCM: H_2_O (2:1), and the final organic phase was separated and evaporated to dryness in a rotary evaporator.

#### 4.1.3. Preparation of the HYNIC-S-(CH_2_)_3_-S-Cholesterol (HYNIC-Cholesterol)

The preparation of HYNIC-Cholesterol was carried out by the reaction between the derivative from SH-(CH_2_)_3_-S-Cholesterol and succinimidyl-N-Boc-HYNIC, with the posterior deprotection of HYNIC, as follows: In a round flask, the SH-(CH_2_)_3_-S-Cholesterol was dissolved in CH_3_CN with constant stirring; posteriorly, NaH was then added in a 1.5:1 stoichiometric ratio concerning the thiolate compound and stirred constantly for at least 15 min. Next, a vial of succinimidyl-N-Boc-HYNIC was taken, resuspended in DCM, and added to the reaction mixture, which was kept under constant stirring at 500 rpm for 3 h. The final product was washed twice with ethanol, followed by distilled water. Finally, the organic phase was evaporated to dryness in a rotary evaporator. The product obtained was Boc-HYNIC-S-(CH_2_)_3_-S-Cholesterol. To eliminate the protective Boc group, the product was dissolved in DCM, and subsequently, a mixture of DCM: TFA 98% was added in a 50:50 *v*/*v* ratio. The reaction mixture was kept under constant stirring for 3 h. The final product was washed twice with ethanol and then with distilled water. The final organic phase obtained was evaporated to dryness in a rotary evaporator.

#### 4.1.4. Preparation of the DOTA-Benzene-p-SC-NH-(CH_2_)_2_-NH-Cholesterol (DOTA-Cholesterol) Derivative

The preparation of DOTA-Cholesterol was performed by the spontaneous reaction of the amino-Cholesterol derivative with DOTA-benzene-p-SCN. Briefly, in a round flask, a certain amount of DOTA-benzene-p-SCN was added together with NH_2_-(CH_2_)_2_-NH-Cholesterol in a 1.5:1 stoichiometric ratio and 5 mL of CH_3_CN. The mixture was kept stirring at room temperature for 2 h. At the end of the reaction, the excess solvent was evaporated by rotary evaporation. Subsequently, the final product was washed with a mixture of DCM: H_2_O (2:1). The mixture was transferred to a separatory funnel, and the organic phase was washed twice with distilled water. This final organic phase was separated and evaporated to dryness on a rotary evaporator.

### 4.2. Preparation of HYNIC-rHDL and DOTA-rHDL

Empty HYNIC-rHDL and DOTA-rHDL nanoparticles and Rhodamine 6G (R6G)-loaded-rHDL were prepared using a modified methodology described previously [57]. The total amount of cholesterol was divided into cholesterol/cholesterol-modified (HYNIC-Cholesterol or DOTA-Cholesterol) in a ratio of 70:30. In a glass vial, the following compounds dissolved in chloroform were mixed: 300 μL of egg yolk phosphatidylcholine (EYPC, 10 mg/mL), 4.9 μL of free cholesterol (FC, 10 mg/mL), 21 μg of cholesterol-modified (HYNIC- or DOTA-), and 7.5 µL of cholesterol ester (CE, 4 mg/mL) dissolved in chloroform. To prepare the R6G-rHDL, Rhodamine was added and incorporated in the oil phase, previously dissolved in chloroform. The mixture was homogenized and vacuum dried in an ultrasonic bath at 4 °C or under a flow of N_2_(g). When all the solvent was removed, a lipid film adhered to the surface of the vial was formed, which was dispersed with 60 μL of DMSO. One milligram of Apo-A1 and 140 μL of sodium cholate (20 mg/mL in buffer) were added, and the total volume was made up to 2 mL with a buffer composed of 10 mM Tris buffer, 0.1 M KCl, and 1 mM adjusted EDTA at pH 8. The vial was closed and homogenized in a vortex. The mixture was transferred to a 14 kDa cellulose dialysis membrane and dialyzed at 4 °C for 48 h with 3–4 changes of the dialyzing solution (buffer solution). Once the dialysis was finished, the content of the membrane was filtered using first a 0.45 μm Millipore filter and then a 0.22 μm filter. The final filtrate was transferred to a dark glass bottle.

### 4.3. Radiolabeling of HYNIC-rHDL and DOTA-rHDL Nanoparticles

#### 4.3.1. Preparation of ^99m^Tc-HYNIC-rHDL Nanoparticles

In a glass vial, 200 μL of the HYNIC-rHDL suspension, 500 μL of EDDA-tricine solution (30 mg of EDDA in 1.5 mL of 0.1 M NaOH and 60 mg of tricine in 1.5 mL of 0.2 M PBS, pH 7.0), 20 μL of SnCl_2_ (1 mg/mL in 0.012 M HCL), and 300 μL ^99m^TcO_4_^-^ (185 MBq) were mixed. The vial was immediately incubated at 40 °C for 30 min. After the incubation, the solution was allowed to reach room temperature, and a sample was taken to determine the reaction yield (labeling) by ITLC-SG using 0.9% NaCl as the mobile phase. The glass vial mixture was transferred to a membrane centrifuge tube (MWCO 100,000 Da) and centrifuged for 15 min at 500× *g*. Finally, the radiochemical purity of ^99m^Tc-HYNIC-rHDL was calculated by ultracentrifugation (^99m^Tc-HYNIC-rHDL remains in the filter while free ^99m^TcO_4_^-^ passes in the filtered solution), ITLC-SG using 0.9% NaCl (Rf = 0 for ^99m^Tc-HYNIC-rHDL and Rf = 1 for free ^99m^TcO_4_Na), and size-exclusion HPLC using a ProteinPak 300SW gel filtration column (Waters) eluting with 0.1 M phosphate pH 7.4 at a flow rate 1.0 mL/min (retention time of nanoparticles 4.7–5.5 min and 13 min for ^99m^TcO_4_Na). Chromatograms were obtained using a NaI(Tl) radiometric detector and a UV-Vis detector. The compound first passed by the UV-Vis detector and after 0.7 min by the radioactive detector. The correspondence of the retention times of the UV-Vis peak and radioactive peak in the chromatogram was considered as a proof of the sample chemical identity. For comparative purposes, HYNIC-Cholesterol was also labeled under the same conditions, except the reaction temperature, which in this case was 95 °C.

#### 4.3.2. Preparation of ^177^Lu-DOTA-rHDL Nanoparticles

In a glass vial, 200 μL of the DOTA-rHDL suspension, 500 μL of 0.2 mol/L acetate buffer, pH 5.0, and 10 μL of a ^177^LuCl_3_ solution (37 MBq) were mixed. The mixture was incubated for 1 h at 37 °C. At the end of the incubation, the reaction yield (labeling) was determined by radio-HPLC. The mixture from the glass vial was transferred to a membrane centrifuge tube (MWCO 100,000 Da) and centrifuged for 15 min at 500× *g*. Finally, the radiochemical purity of ^177^Lu-DOTA-rHDL was calculated by ultracentrifugation (^177^Lu-DOTA-rHDL remained in the filter, while free ^177^LuCl_3_ passed in the filtered solution) and size-exclusion HPLC using a ProteinPak 300SW gel filtration column (Waters) eluting with 0.1 M phosphate pH 7.4 at a flow rate 1.0 mL/min (retention time of nanoparticles 4.7–5.5 min and 12.5 min for ^177^LuCl_3_). For comparative purposes, DOTA-Cholesterol was also labeled under the same conditions, except the reaction temperature, which in this case was 95 °C.

### 4.4. Characterization and Biological Evaluation

#### 4.4.1. Physicochemical Characterization of Cholesterol Derivatives, HYNIC-rHDL and DOTA-rHDL

Dehydrated samples of the products for each reaction step: Br-Cholesterol, SH-(CH_2_)_3_-S-Cholesterol, HYNIC-Cholesterol, Cl-Cholesterol, NH_2_-(CH_2_)_2_-NH-Cholesterol and DOTA-Cholesterol, were physicochemical characterized using UV-Vis (Thermo Genesys 10S, resolution 0.5 nm, operating in a range of 200–800 nm) and FT-IR spectroscopy (Perkin Elmer System 2000 with ATR (Pike Technologies), resolution 0.4 cm^−1^, 40 scans, and operating in a range of 4000–400 cm^−1^); the melting point and Rf by ITLC were also obtained.

HYNIC-rHDL and DOTA-rHDL nanoparticles were characterized by DLS (Nanotrac wave analyzer) to obtain the NPs’ size, polydispersity, and stability, which were evaluated using the Zeta potential at 0, 2, 4, 6, 12, and 24 h after preparation.

#### 4.4.2. In Vitro and In Vivo Evaluation of HYNIC-rHDL and DOTA-rHDL Nanoparticles

##### In Vitro Uptake of Rhodamine 6G Carried in HYNIC-rHDL and DOTA-rHDL

Here, 2.5 × 10^5^ cells T47D cells (SR-B1+) were seeded on 7 chamber slides and incubated in 5% CO_2_ for 24 h. At the end of the incubation, the following treatments were applied (one per slide): (i) 30 μL of medium (control), (ii) 30 μL of HYNIC-rHDL/R6G, (iii) 30 μL of HYNIC-rHDL/R6G after 30 min pre-treatment with 100 μL Apo-A1 (1 mg/mL) (in the form of empty rHDL), (iv) 30 μL HYNIC-rHDL/R6G after 30 min pre-treatment with 200 μL Apo-A1 (1 mg/mL) (in the form of empty rHDL), (v) 30 μL DOTA-rHDL/R6G, (vi) 30 μL DOTA-rHDL/R6G after 30 min pre-treatment with 100 μL Apo-A1 (1 mg/mL) (in the form of empty rHDL), (vii) 30 μL DOTA-rHDL/R6G after 30 min pre-treatment with 200 μL Apo-A1 (1 mg/mL) (in the form of empty rHDL). In all samples, the cells were incubated 1 h after the addition of the treatment. After this time, the chambers were turned over to remove the medium. They were washed three times with 1X PBS (pH 7.4) and then fixed with 2% of formaldehyde for 15 min. After this time, they were washed again with 1X PBS. The fluorescent dye 4′,6-diamidino-2-phenylindole (DAPI) was added to all samples, and they were covered with a coverslip. They were incubated at 37 °C for 5 min in the dark before image acquisition in the Nikon eclipse fluorescence microscope.

##### In Vitro Stability and Cellular Uptake of ^99m^Tc-HYNIC-rHDL and ^177^Lu-DOTA-rHDL

^99m^Tc-HYNIC-rHDL and ^177^Lu-DOTA-rHDL. (3.7 MBq, 50 µL) were incubated at 37 °C with 1 mL of 0.9% NaCl or 1 mL of human serum (IST-909c, Sigma-Aldrich). Samples of the mixtures were analyzed after 24 h for size-exclusion radio-HPLC analysis using a ProteinPak 300SW Waters column and 0.01M PBS as the mobile phase (1 mL/min). In this system, the retention time of nanoparticles was 4.7–5.5 min, 12.5 min for free ^177^Lu (^177^LuCl_3_), and 13 min for free ^99m^Tc (^99m^TcO_4_Na).

In a 48-well plate, 20 of them were seeded with PC3 cells (SR-B1+). In each well, 5.0 × 10^4^ cells were deposited in 250 µL of RPMI medium supplemented with 10% fetal bovine serum and antibiotics (100 µg/mL penicillin and 100 µg/mL streptomycin). Cells were incubated at 37 °C and 5% CO_2_ for 24 h. Next, the following treatments were added (*n* = 5) with an incubation time of 1h: (i) 30 μL of ^99m^Tc-HYNIC-rHDL, (ii) 30 μL of ^99m^Tc-HYNIC-rHDL previously treated for 30 min with 100 μL of Apo-A1 (1 mg/mL) (in the form of empty rHDL), (iii) 30 μL of ^177^Lu-DOTA-rHDL, (iv) 30 μL of ^177^Lu-DOTA-rHDL previously treated for 30 min with 100 μL of Apo-A1 (1 mg/mL) (in the form of empty rHDL). The total activity added to the wells was previously calculated (*n* = 3 for each) for 30 μL of ^99m^Tc-HYNIC-rHDL and 30 μL of ^177^Lu-DOTA-rHDL, respectively, using a crystal scintillation well-type detector (Auto In-v-tron 4010; NML Inc., Houston, TX, USA). After incubation, the supernatant of each well was removed and measured in the crystal scintillation well-type detector. Then, the wells were washed 2 times with 250 μL of PBX 1X (pH 7.4), and the radioactivity of the washes was also measured. The sum of the activity of the treatment removed and of the two washes constituted the activity not taken up by the cells. After washing the cells, they were incubated twice for 5 min with 250 μL of glycine-HCl (50 mM, pH 2.8). Supernatants from the two glycine-HCl treatments were pooled, and the activity was measured. The activity found corresponds to that adsorbed or bound to the cell membrane. The cells were then washed with 1X PBS (pH 7.4) and incubated twice with 250 μL NaOH (1 M) for 5 min. The supernatants were again combined and measured. The activity found in this fraction corresponds to that internalized (cytosol) in the cell. The results are expressed as % cellular uptake and % internalization (cytosol accumulation) and were calculated and compared with previously reported work [7].

#### 4.4.3. In Vivo Evaluation of ^99m^Tc-HYNIC-rHDL and ^177^Lu-DOTA-rHDL

Healthy CD1 male mice weighing 30–35 g were injected intravenously with ^99m^Tc-HYNIC-rHDL (200 µL, 3 MBq) or ^177^Lu-DOTA-rHDL (100 µL, 2 MBq) under isoflurane (2%) anesthesia. Mice were sacrificed and dissected at 2 and 24 h after injection (*n* = 3 for each nanosystem). The heart, lungs, liver, spleen, kidneys, and pancreas were extracted, and their activity was measured on a NaI(Tl) detector. The measured activity in each organ was corrected by radioactivity decay using a standard, and the average activity represents the percentage of activity injected per organ. Whole-body molecular images were obtained from six mice injected with the radio-nanosystems via a radioisotopic/X-ray image (^99m^Tc-HYNIC-rHDL, *n* = 3) and an optical image (Cerenkov/X-ray image) (^177^Lu-DOTA-rHDL, *n* = 3) using a preclinical imaging system equipped with a CCD camera (XTREME Imaging System; Bruker, Billerica, MA, USA).

#### 4.4.4. Statistical Analysis

To compare the results of the experiments, the t-Student test (*n* = 3 or *n* = 5, depending on the experiment) was used. Differences with a *p* < 0.05 for the two-tailed or one-tailed test, according to the experiment, were considered significant.

## 5. Conclusions

Two cholesterol derivatives were successfully used for the synthesis of surface-modified rHDL (HYNIC-rHDL and DOTA-rHDL), capable of complexing radionuclides for SR-B1 molecular imaging. HYNIC-rHDL can be used for ^99m^Tc SPECT imaging, while DOTA-rHDL can complex a wide variety of radionuclides for SPECT and PET imaging (e.g., ^111^In, ^177^Lu, ^64^Cu, and ^68^Ga,). Through the encapsulation of Rhodamine 6G in the lipidic core of HYNIC-rHDL and DOTA-rHDL, the ability of the nanoparticles to internalize and release cargos (e.g., therapeutic agents) into the cytosol of cancer cells with high expression of SR-B1 was demonstrated. Therefore, HYNIC-rHDL and DOTA-rHDL are versatile prototype platforms for the development of theranostic nanosystems since they can carry a great variety of therapeutic agents into the core while providing non-invasive SR-B1 molecular imaging. Of particular interest is the biodistribution profile of DOTA-rHDL due to its lower hydrophobicity with regard to the unmodified rHDL.

## 6. Patents

The patent application was filed in Mexico, No. MX/E/2021/087172 “Nanoplataforma prototipo DOTA-rHDL con hidrofilicidad mejorada para la preparación de nanosistemas teranósticos”.

## Figures and Tables

**Figure 1 molecules-27-07046-f001:**
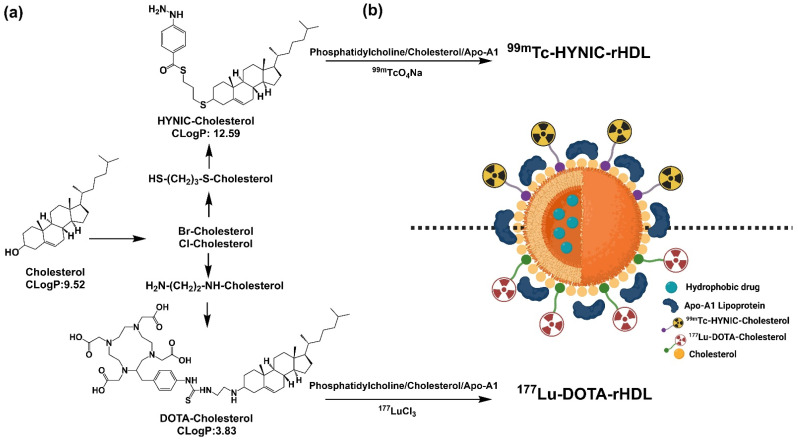
Schematic methodology for the preparation of a suitable platform for theranostic applications. (**a**) Synthesis of cholesterol derivatives; (**b**) HYNIC-rHDL or DOTA-rHDL nanoparticles radiolabeled with ^99m^Tc and ^177^Lu.

**Figure 2 molecules-27-07046-f002:**
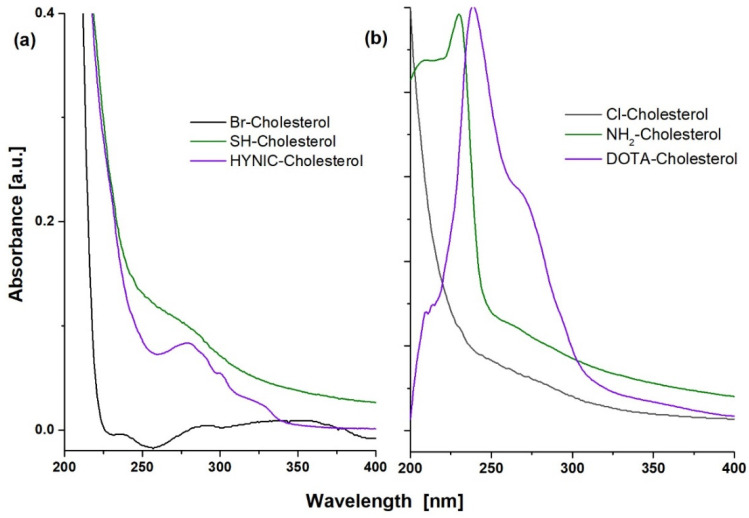
UV-Vis spectra of (**a**) HYNIC-S-(CH_2_)_3_-S-Cholesterol (HYNIC-Cholesterol) and its Br- and SH- precursors and (**b**) DOTA-benzene-*p*-SC-NH-(CH_2_)_3_-NH-Cholesterol (DOTA-Cholesterol) and its Cl- and NH_2_- precursors.

**Figure 3 molecules-27-07046-f003:**
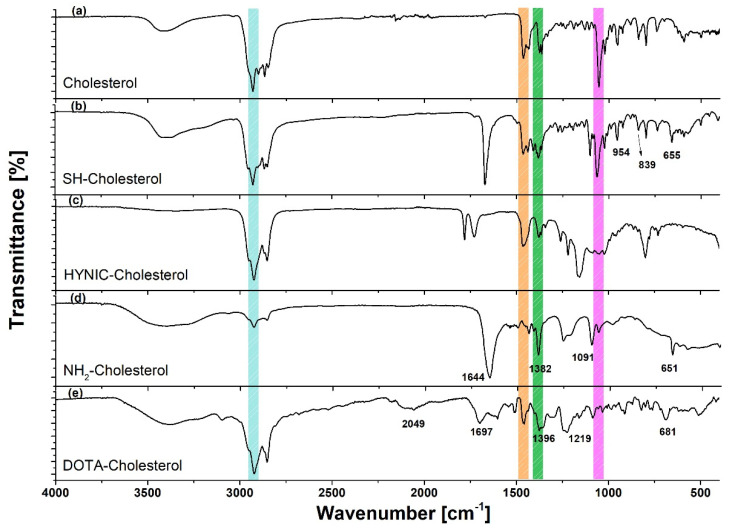
Infrared spectra of cholesterol and derivatives: raw material (cholesterol), SH-(CH_2_)_3_-S-Cholesterol (SH-Cholesterol), HYNIC-S-(CH_2_)_3_-S-Cholesterol (HYNIC-Cholesterol), NH_2_-(CH_2_)_2_-NH-Cholesterol (NH_2_-Cholesterol), and DOTA-benzene-*p*-SC-NH-(CH_2_)_3_-NH-Cholesterol (DOTA-Cholesterol).

**Figure 4 molecules-27-07046-f004:**
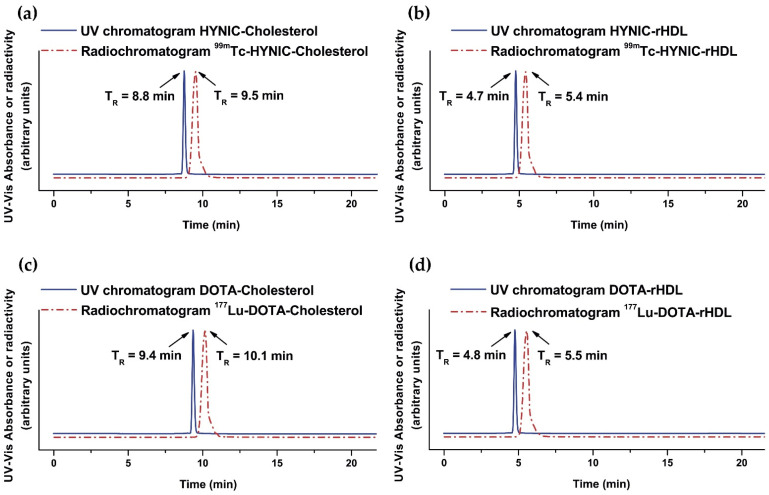
Size-exclusion HPLC chromatogram (254 nm; blue continuous line) of (**a**) HYNIC-Cholesterol, (**b**) HYNIC-rHDL nanoparticles, (**c**) DOTA-Cholesterol, and (**d**) DOTA-rHDL nanoparticles. Size-exclusion HPLC radiochromatogram (red dotted line) of (**a**) ^99m^Tc-HYNIC-Cholesterol, (**b**) ^99m^Tc-HYNIC-rHDL nanoparticles, (**c**) ^177^Lu-DOTA-Cholesterol, and (**d**) ^177^Lu-DOTA-rHDL nanoparticles. Protein-Pak 300SW column, 0.1 M phosphate buffer (pH 7.4), flow rate 1.0 mL/min.

**Figure 5 molecules-27-07046-f005:**
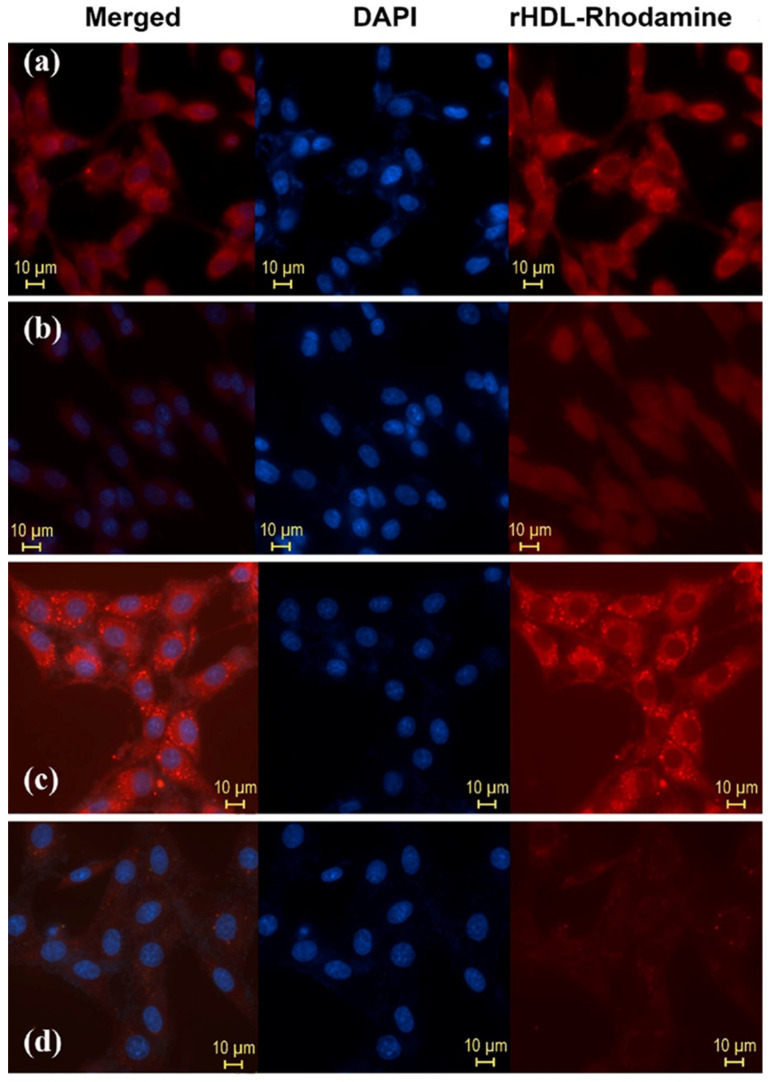
Internalization in T47D cells (SR-B1+) (release to cytosol) of Rhodamine 6G used as cargo in (**a**) HYNIC-rHDL nanoparticles, (**b**) HYNIC-rHDL nanoparticles with SR-B1 receptor blocking effect (T47D cells pretreated with 200 μL of Apo-A1, 1 mg/mL, in the form of empty rHDL), (**c**) DOTA-rHDL nanoparticles, and (**d**) DOTA-rHDL nanoparticles with SR-B1 receptor blocking effect (T47D cells pretreated with 200 μL of Apo-A1, 1 mg/mL, in the form of empty rHDL). Observe the intensity of Rhodamine fluorescence when it is internalized in TD47 cells due to its release from the rHDL core to the cytosol after the specific interaction of Apo-A1 (present on the surface of the rHDL nanoparticle) with SR-B1 (highly expressed on the surface of TD47 cancer cells).

**Figure 6 molecules-27-07046-f006:**
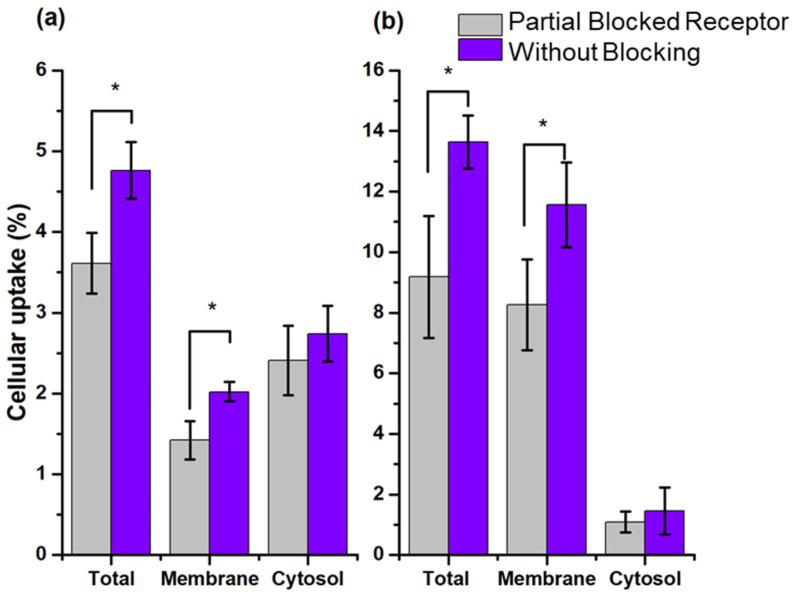
Cellular uptake and internalization (cytosol accumulation) (% of total activity) of (**a**) ^99m^Tc-HYNIC-Cholesterol and (**b**) ^177^Lu-DOTA-Cholesterol in PC3 with blocked and unblocked SR-B1. Blocking of SR-B1 was performed with 7 mg/mL of empty rHDLs. * Statistically significant difference (*p* < 0.05).

**Figure 7 molecules-27-07046-f007:**
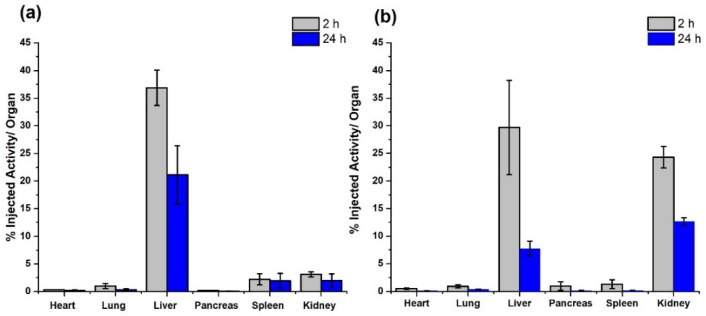
Biodistribution profile in mice (% injected activity/organ) of (**a**) HYNIC-rHDL and (**b**) DOTA-rHDL nanoparticles after 2 and 24 h of administration.

**Figure 8 molecules-27-07046-f008:**
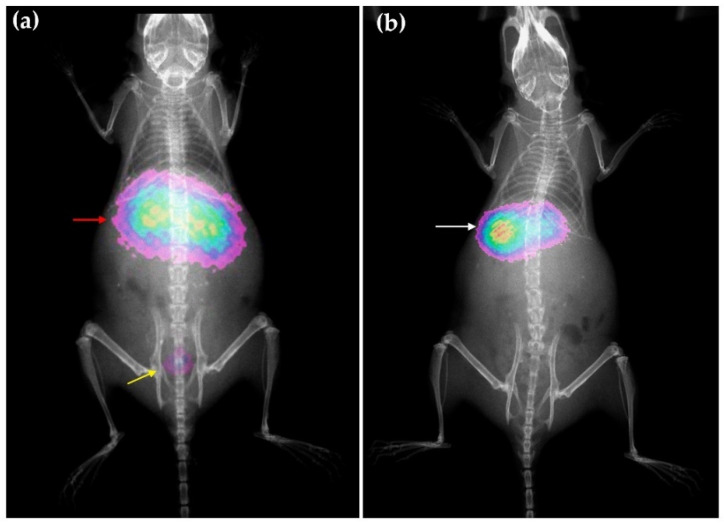
Molecular imaging obtained in mice at 2 h after intravenous administration of rHDL nanosystems, (**a**) optical image (Cerenkov/X-ray image) for ^177^LuDOTA-rHDL and (**b**) radioisotopic/X-ray image for ^99m^Tc-HYNIC-rHDL. Observe the overlapping of liver and kidney (red arrow) as a consequence of the ^177^Lu-DOTA-rHDL uptake in both organs, as well as the radiotracer elimination through the bladder (yellow arrow); ^99m^Tc-HYNIC-rHDL showed mainly liver uptake (white arrow).

**Table 1 molecules-27-07046-t001:** Physicochemical properties of cholesterol derivatives.

Product	Yield	Physical Appearance	Melting Point	Rf (TLC-SG)
Cholesterol	--	White fine powder, oily texture	148–150	0.60
Br-Cholesterol	30–40	Off-white light-yellow powder	135–137	0.57
Cl-Cholesterol	30–40	White-yellowish powder, oily texture.	95–105	0.45
SH-(CH_2_)_3_-SH-Cholesterol	65–75	Off-white light-yellow powder	139–142	0.74
NH_2_-(CH_2_)_2_-NH-Cholesterol	60–70	Bright yellow, hygroscopic conglomerate	121–130	0.83
HYNIC-S-(CH_2_)_3_-S-Cholesterol	70–80	Off-white light-yellow powder	115	0.67
DOTA-benzene-*p*-SC-NH-(CH_2_)_2_-NH-Cholesterol	<90	Fine cream-colored powder with an oily texture	133	0.83

## Data Availability

Not applicable.

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
