# Peer review of "Engineered rHDL Nanoparticles as a Suitable Platform for Theranostic Applications"

_molecules, 2022, doi:10.3390/molecules27207046_

Round 1
Reviewer 1 Report
The work presented in the manuscript submitted by Aranda-Lara et al. is original, with a high degree of novelty, which can arouse the interest of the researchers in the field. The manuscript is well organized and written and the conclusions are supported by the reported results.
It is my opinion that the manuscript can be accepted in the present form.
Author Response
Dr. Kristina Djanashvili
Dr. Sara Lacerda
Guest Editors
Molecules
Manuscript ID: molecules-1898185
Title: Engineered of modified rHDL nanoparticles as suitable platform for theranostic applications
Dear Professors,
Answer: The authors appreciate the reviewer's comments.

Reviewer 2 Report
The work presents the preparation of nanoparticles of HYNIC-rHDL and DOTA-rHDL to vehiculize radionuclides for potential theragnostic applications.
The topic is interesting and comprises many experimental techniques and results. However, the presentation of the results is confusing and the following questions arise:
1) Synthesis of cholesterol derivatives is only explained in a short paragraph in page 8 lines 247-255. Writing of this paragraph is confusing and no scheme of the synthesis method is included which makes it difficult for the reader to understand which are the intermediate products that appear in Table 1. Furthermore, in the discussion Figure 1 a, b and c are mentioned but this Figure is not included in the manuscript
2) Table 1 contains the RF in iTLC -SG of different species. Which is the utility of these data, since they are not mentioned in the results or discussion.
3)Labelling yield was 55% for 99mTc and 85% for 177Lu. Which are the impurities? Can all of them be determined by iTLC chromatography? Which are the Rf of the labelled product and of the expected impurities in the system used for quality control?
4)The selection of the cell lines used in the experiments is not justified? Why were different types of cells used for the fluorescence experiments and the cell experiments?
5)Blocking experiments were also performed using different blocking agents? Why?
6) In page 5 line 153-155 the authors claim that “the size of all samples is adequate to avoid phagocytosis by the RES and enter the tumour microenvironment through the fenestrated microvasculature” Please include references.
7) In page 8 lines 239-242 the authors wrote: “we hypothesize that, if rHDL is prepared with an amphiphilic derivative of cholesterol synthesized with a bifunctional agent, the hydrophobic domine of the molecule would be "embedded" inside the lipidic nucleus, while the hydrophilic part (which includes the chelating agent) would be exposed to the outside, then radio labelling can be carried out.” How was this hypothesis proven?
8) The ability for loading and delivery was studied using Rhodamine 6G as a model of therapeutic drug. Is this ability not modified by the physicochemical properties of the therapeutic drug? Is the use of Rhodamine 6G a standard procedure to study loading of NP? Please include references
9) Table 2 contains the results of uptake in PC3 cells expressed as uptake ratios (blocked cells/unblocked cells). This form of expressing the results is confusing. Wouldn´t be better to express them either as simple % of uptakes in the different conditions or us % of specific uptake (unblocked-blocked/ unblocked)?
10) Also the form of expressing the results in Figure 5 as a ratio with respect to previously reported rHDL with the radionuclide inside is confusing as it is also the discussion of these results. For example, the authors claim that “that DOTA conjugation (5b.) induced greater behavioural changes in the biodistribution with respect to rHDL modified with HYNIC (5a.).”. However, in my understanding a ratio of aprox. 1 means that the results are the same and, in the graph, DOTA-rHDL labelled with 177Lu has more organs with a ratio around 1 than Hynic-rHDL
Author Response
Dr. Kristina Djanashvili
Dr. Sara Lacerda
Guest Editors
Molecules
Manuscript ID: molecules-1898185
Title: Engineered of modified rHDL nanoparticles as suitable platform for theranostic applications
Dear Professors,
Please find attached the answers, point-by-point, to the reviewer' comments.

Reviewer 3 Report
Manuscript entitled “Engineered of modified rHDL nanoparticles as suitable plat- 2 form for theranostic applications” submitted by Liliana A-L et al. describes the preparation of modified rHDL nanoparticles, characterization of HYNIC-rHDL and DOTA-Rhdl, their in vitro and in vivo studies. Please see comments below.
11) The authors need to mention the bifunctional agents used from their work. It should be clear for the reader to thoroughly follow the manuscript.
22) There are typos throughout the manuscript. For example, line 43 and line 2, line 109 and so on. Please proofread the manuscript. In line 241, please correct the typo as well.
33) In the lines 100, 116, the references or weblinks need to be included in the references section but not in the main text. Please move them.
44) It is reported that the authors have used both UV-Vis and IR to characterize the Cholesterol derivatives (fig.3) but it would be ideal to include their NMR and HPLC (purity) data for confirmation of characterization. Please include them.
55) It is not explained why the authors have seen more uptake in intestine in both cases of HYNIC-rHDL and DOTA-rHDL nanoparticles in their discussion. Please give a rationale. How sensitive these nanoparticles under different PH conditions? Please test the stability of radiolabeled nanoparticles of both and include.
66) Please label fig 5 as a and b respectively.
77) In section 2.4.3, Please include any SPECT images, if any, of HYNIC-rHDL and DOTA-rHDL nanoparticles in in vivo evaluation.
Please consider as a major revisions and address them.
Author Response

(The authors gave the same response as above.)

Round 2
Reviewer 2 Report
The manuscript has been considerably improved and is now adequate for publishing
Author Response
The manuscript has been considerably improved and is now adequate for publishing
Answer: The authors appreciate the reviewer's comments, which led to changes in the revised manuscript to substantially enrich and improve it.
Reviewer 3 Report
The authors have provided and revised their manuscript to some extent and please accept.
Author Response
The authors have provided and revised their manuscript to some extent and please accept.
Answer: The authors appreciate the reviewer's comments, which led to changes in the revised manuscript to substantially enrich and improve it.
Sincerely yours,
Enrique Morales-Avila, Ph.D.